# Neural responses to shot changes by cut in cinematographic editing: An EEG (ERD/ERS) study

Javier Sanz-Aznar[1]*, Lydia Sánchez-Gómez[1], Luis Emilio Bruni[2], Carlos Aguilar-Paredes[1], Andreas Wulff-Abramsson[2]

1 Information and Audiovisual Media, University of Barcelona, Barcelona, Spain, 2 Media Technology, Aalborg University, Copenhagen, Denmark

* javier.sanz@ub.edu

**Data Availability Statement:** All files are available from the Open Science Framework database (https://osf.io/exbt7/).

## Abstract

In order to analyze and detect neural activations and inhibitions in film spectators to shot changes by cut in films, we developed a methodology based on comparisons of recorded EEG signals and analyzed the event-related desynchronization/synchronization (ERD/ERS). The aim of the research is isolating these neuronal responses from other visual and auditory features that covary with film editing. This system of comparing pairs of signals using permutation tests, the Spearman correlation, and slope analysis is implemented in an automated way through sliding windows, analyzing all the registered electrodes signals at all the frequency bands defined. Through this methodology, we are able to locate, identify, and quantify the variations in neuronal rhythms in specific cortical areas and frequency ranges with temporal precision. Our results detected that after a cut there is a synchronization in theta rhythms during the first 188 ms with left lateralization, and also a desynchronization between 250 ms and 750 ms in the delta frequency band. The cortical area where most of these neuronal responses are detected in both cases is the parietal area.

## Introduction

Neurocinematics represents a new way to approach cinematographic theory [1], leading to a paradigm shift in the methodology of cinematographic studies. Based on ecological cognitive film theory [2], neurocinematics shifts the central focus from the film itself towards the cognitive system of the spectator watching the film, evaluating the spectator's neuronal responses to the different stimuli present in the film. In this theoretical line, researchers like Zacks [3] propose that the perceptual system does not differentiate between real and filmic stimuli, as both are processed in the same way. Carroll and Seeley [4] even suggest that cognitive mechanisms are more effective at processing cinematographic stimuli than real stimuli, because the film offers a clear, simplified representation of the facts, which they describe as *uncluttered clarity*.

From the neurocinematic point of view, the shot change by cut has been studied by means of various biometric measurements, introducing new approaches to cinematographic theory. Based on eye-blinking, eye movements, and the cognitive limitations of change blindness [5]

**Funding:** The authors received no specific funding for this work.

**Competing interests:** The authors have declared that no competing interests exist.

and inattentional blindness [6], Smith [7] proposed an attentional theory of cinematic continuity to explain how the spectator perceives the event of the cut, introducing the concept of *edit blindness* [8]. Change blindness and inattentional blindness have been posited as two situations in which our cognitive system does not perceive visual changes.

Smith justifies and explains the classical editing rules—established over years of cinematographic praxis [9–11]—based on the relevant mechanisms of the cognitive system. In a magnetic resonance study, Ben-Yakov and Henson [12] identified hippocampal activation during shot changes, linking the cuts to the management of short-term and long-term memory, as well as to semantic memory. Ben-Yakov and Henson's research points back to Mitry's [13] theoretical perspective on editing, which defines the cut as an intellectual and discursive process involving encoding and memory processes, in contrast with the emotional nature of the shot. In the same vein of the aforementioned study on shot changes by editing, other approaches making use of EEG have been able to differentiate between *related* and *unrelated cuts* [14], and also between cuts crossing the axis and cuts keeping the 180º axis [15]. All the research referred to above confirms that the methodology applied in neurocinematics is an effective way of approaching film phenomena to investigate the intricacies of the shot change by cut, in the interests of unravelling the nature of editing and its relationship with the spectator's cognitive processes.

In the interests of contributing to our understanding of the nature of the cinematographic cut and the cognitive processes that it triggers, the objective of this research is to characterize and identify relevant electroencephalogram patterns in the neuronal responses of spectators exposed to shot changes introduced by continuity editing cuts. This study should reveal some of the brain correlates involved in making sense of the cuts, thus allowing us to assess the cognitive implications of the reaction to the cut.

## Methodology

In order to locate and identify neuronal patterns triggered by cut events, we designed an experiment for recording the EEG while subjects observed different film scenes containing shot changes introduced by cuts. Once recorded, the signals were prepared for analysis of the frequency domain, specifically, power changes. We then performed a statistical analysis of the power of all the signals in order to locate through automated computer processes all the response patterns produced as a result of the shot change by cut. Once these three phases were completed, the significant time windows located were analyzed in terms of event-related desynchronization/synchronization (ERD/ERS). The ERD/ERS is an analysis of the EEG signal in the frequency domain that allows us to analyze neuronal excitations and inhibitions.

In the present investigation, we are interested in locating those neuronal areas that show neuronal activation or inhibition, trying to establish a map of those neuronal areas involved in the processing of the shot change by cut. The analysis in the frequency domain allows us to differentiate, depending on the oscillation of the analyzed wave, the degree of neuronal excitation that is taking place at each moment, allowing us to establish a pattern of neuronal activations and inhibitions in reaction to the shot change by cut.

### Film excerpts

The experimental design consisted of the selection of four film excerpts from commercial feature films, and the design of the electroencephalographic record. The film excerpts were complete scenes containing shot changes produced by cuts with continuity editing. For this purpose, we selected real audiovisual references rather than our own *ad hoc* materials, as has been customary in previous studies with similar characteristics [15, 16]. We selected four film

excerpts from different eras, with different techniques and cinematographic styles, in order to obtain the widest possible representation of reactions related to technical and aesthetic aspects, which was constant in each excerpt selected. The objective of this criterion was to be able to differentiate the spectator's neuronal responses to the cuts from those due to the aesthetic, stylistic, or technical characteristics of the film.

The film excerpts selected were classified as fiction films [17] and live action films [18]. As the research interest is to study the predominant film mode of representation, we favoured excerpts from films that used the IMR (institutional mode of representation) defined by Burch [19], while dismissing cases of PMR (primitive mode of representation) and AMR (alternative mode of representation). The main reason for excluding films in the PMR category is that the shot change by cut in such films does not have a narrative function within the scene but is used out of technical necessity to join one scene to the next, or simply as a tool to modify the physical duration of the negative by patching together several rolls of film through editing. On the other hand, films in the AMR category were excluded because they generally seek to challenge the institutional mode of representation by exploring new forms of expression afforded by cinematographic techniques, in many cases based on narrative abstraction. We also excluded all silent film production, based on the understanding that the IMR only really took shape with the inclusion of synchronous soundtracks in films [20].

Unlike Francuz and Zabielska-Mendyk [14], whose aim was to detect EEG differences between cuts based on discontinuity (unrelated) from those based on continuity (related), we focused on cuts with space and time continuity in the same scene, given the fact that this is the kind of shot change that the spectator processes more naturally [8]. Due to this characteristic of the shot change, all the cuts analyzed in this article can be classified as *raccord* editing according to Amiel [11]. The temporal relationship between shots around the cut could be defined according to the categories of absolutely continuous cut or *hiatus* according to Burch's classification [21].

The film excerpts selected come from *Bonnie & Clyde* [22], *The Searchers* [23], *Whiplash* [24], and *On the Waterfront* [25]. The excerpts were all played in their original language, without subtitles that could potentially influence the type of specific attention triggered by the film [26, 27]. Considering that the experiment was carried out at the University of Aalborg in Copenhagen, with a high rate of international students who have English as their working language, films in English were chosen. The excerpts were presented to the subjects in random order so that the viewing order would not be a conditioning factor.

## Participants

Twenty-one subjects (twelve male and nine female) participated as spectators, all of whom were from the University of Alborg (mostly undergraduate, master's, or PhD students). The average of the participants age is 26 years, being 38 the oldest one and 22 the youngest one.

The participants were duly informed about the characteristics of the study (without providing any details that could condition their perception) and were requested not to consume stimulants such as coffee or depressants such as alcoholic drinks in the hours prior to the experiment. None of the subjects reported any neurological disorder, psychological problems, or the use of medication. All the subjects accepted and signed the conditions for data collection and anonymous use for non-profit research purposes.

## EEG recording

The EEG signal of the 21 volunteers was recorded using an array of 31 electrodes distributed over the scalp with an additional electrode situated on the orbicular muscle of the left eye. The

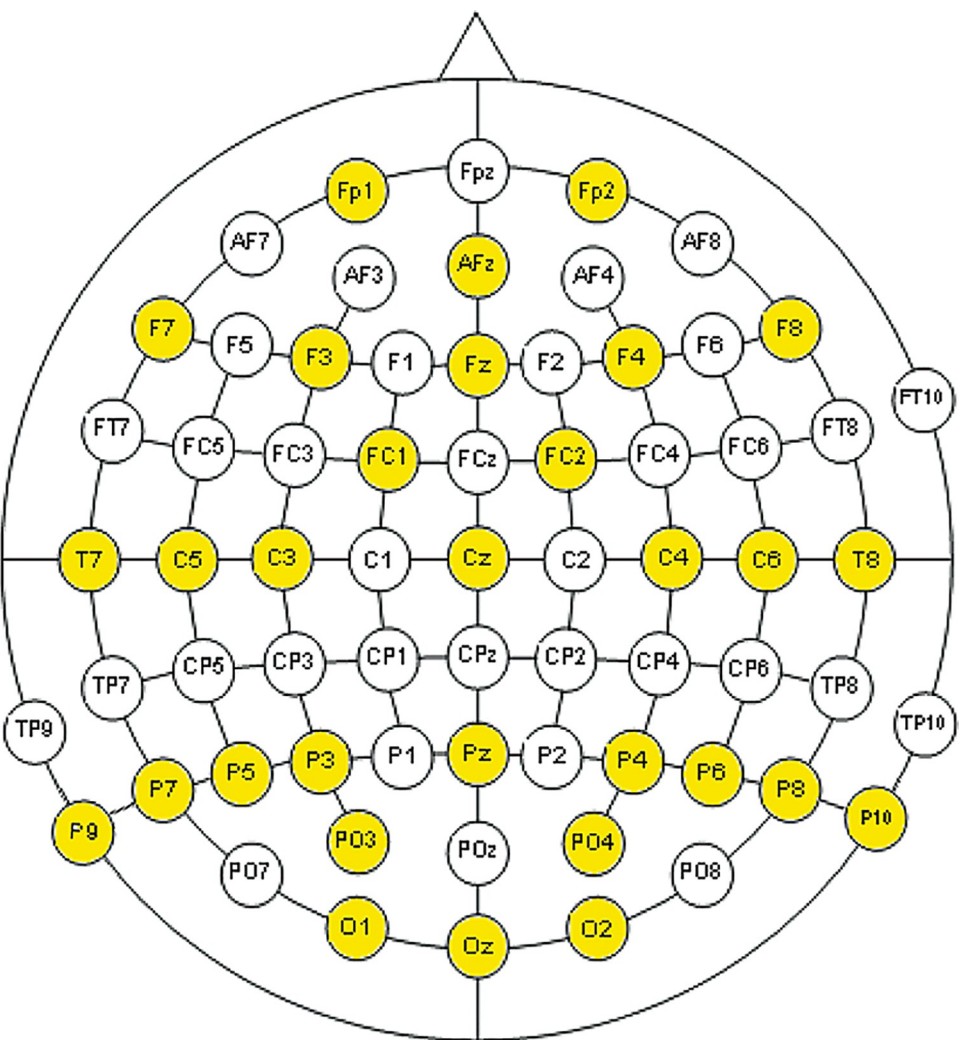

**Fig 1. Distribution of the electrodes following the international 10–20 system for EEG-MNC.**

signal amplifier devices were two g.Tec g.Gammabox and two g.Tec g.USB Amp, with 16 channels each [28]. The amplifier devices were connected using the master-slave connectivity model, resulting in 32 recording channels. The electroencephalogram sampling rate was 256Hz. From the 32 channels, 31 were dedicated to covering different brain areas (Fig 1). The reference electrode was placed in position FP1. The placement of the electrodes follows positions defined by the international convention of the American Electroencephalographic Society in its 10–20 system for EEG-MNC [29].

The electroencephalographic recordings were made between 15 and 30 May 2017. The four film excepts were played to each subject via the Unity game engine, while the electroencephalogram was recorded in MatLab. For operational reasons, Unity and MatLab were run on different synchronized computers via a User Datagram Protocol (UDP) Network.

We realize the EEG experiment under the ethical standards of the Augmented Cognition Lab of the Technical Faculty of IT and Design, at Aalborg University in Copenhagen (Denmark). As proof of it, Aalborg University expended us an ethical approval signed letter with ID 2020-020-00504. All the subjects who participated in the experiment signed a consentient document before initiating the experiment and their EEG recorders was anonymized.

## Signal preparation

After recording the data, an independent component analysis (ICA) was performed in order to isolate the signal for each electrode from its recording area, eliminating interferences from other areas [30]. Then, we performed a manual cleaning of artifacts from the recorded signals.

Once we had the continuous signal cleaned for each electrode from the 21 subjects, we then obtained model signals for each audiovisual clip. We split up the continuous signal from each subject into four fragments, each one corresponding to the viewing of each film excerpt. Then we averaged the 21 signals from each electrode for each of the four excerpts. In this step, for each electrode we got four model signals, each from a film excerpt. This process helped us because the signal epochs that did not show a common neural response pattern in the 21 users while viewing the film were not subsequently reflected as ERD/ERS processes, because the uneven reactions of the users were compensated in the averaging process.

Of the model signals for each audiovisual clip, we were interested in the epochs around each shot change. We therefore focused on the second before the cut, which was used as a reference to identify the change triggered by the cut, and the second after the cut, where we analyzed the reactions to the shot change. From each model signal for each audiovisual clip, we selected 2000ms around each cut and discarded the rest of the signal. In this way, we obtained 84 two-second epochs for *Bonnie and Clyde*, 15 for *The Searchers*, 112 for *Whiplash*, and 47 for *On the Waterfront*. The first second before the cut is used as a control condition signal, allowing us to recognize if the results obtained are really consequence of the event of the cut.

Through the same process as adopted previously, from the isolated signal epochs for each audiovisual clip we obtained a cut model signal for each audiovisual clip and electrode, which we called the ASF (average signal per film). In this way, we strengthened those parts of the signal that remained constant in all the cuts of each film. At the end of this signal process, for each electrode we obtained four 2-second signals, one for each film excerpt viewed, with the first second corresponding to the response before the cut and the next second to the response after the shot change.

We transformed the ASFs into the frequency domain and separated them into different frequency ranges: 0.5-3Hz for delta, 3-7Hz for theta, 7-14Hz for alpha, 14-32Hz for beta, 32-42Hz for gamma, 0.5–1.5Hz for low delta, 1.5-3Hz for high delta, 3-5Hz for low theta, 5-7Hz for high theta, 7–10.5Hz for low alpha, 10.5-14Hz for high alpha 14-23Hz for low beta, 23-32Hz for high beta, 32-37Hz for low gamma and 37-42Hz for high gamma.

To transform the recorded EEG signal to the frequency domain, it is necessary to divide the signal into time intervals that collect each interval the waveform to be analyzed. Therefore, we take sets of 16 successive EEG samples for each signal belonging to each electrode as interval, where apply the fast Fourier transform. Taking into consideration that the EEG has registered 256 samples per second, we obtain a total of 16 samples for each second in the frequency domain, assuming each frequency sample a time interval of 62.5 milliseconds [31]. This process to obtain the ASFs is explained in more detail in a previous publication [32].

## Treatment of the data obtained in the ASFs

Once we obtained the four ASFs broken down into 15 frequency ranges, we conducted a computationally automated comparative analysis that allowed us to locate all those time intervals where there were neuronal responses to the shot change by cut. This step is necessary since we are using film fragments, allowing us to locate those time windows that reflect neuronal responses exclusively triggered by the shot change by cut event. In this way, we locate the time interval, electrode and frequency where we have to apply the study of ERD/ERS.

To ensure the temporal precision of the results obtained, our signal comparison system was applied using sliding time windows with a six-sample dimension, so we had 27 time windows (13 time windows before the cut and another 14 after). The time window number 14 was comprised by the three samples immediately before the cut and the three samples immediately after the cut, representing the temporal window of the signal centred on the event of the cut. Each time window contained in an electrode for a specific frequency range was compared with its equivalent signal from different films. In this way, we applied a pairwise comparison that offered six possibilities, because we have ASFs from four films for each electrode in each frequency range. In the pairwise comparisons we looked for signal intervals that showed dependence and correlation in order to confirm that it reflected a neuronal response to the only common event, the shot change by cut. To this end, we applied the Spearman test and the permutation test in each pairwise comparison. Due to the high number of comparisons performed, it is necessary to avoid the possibility of false positives. The combined use of the permutations test and the Spearman correlation test allow obtaining robust results [33], avoiding type I errors or false positives [34, 35]. The joint use of both tests allows to eliminate false positives because it allows to discard all those results that reflect correlation but do not show dependency, at the same time that we eliminate all those that, even showing dependency, do not show correlation.

If the six possible comparisons showed significant values of dependence (p-value <0.05) and correlation (Rho> 0.5), it would be considered that this time window, for that electrode and in that frequency range, reflected a neuronal response triggered by the shot change by cut. This analytic process is explained in detail in a previous publication dedicated exclusively to expose the signal analysis methodology applied [32].

As a result of this process of correlation and dependence between signals we created a matrix called CD (correlation and dependence) for each frequency range $j$ ($\forall\, j \in [1,15]$), as shown in Eq 1. Each row of the matrix represents an electrode $E_k$ ($\forall\, k \in [1,31]$) and each column represents a time window $VT_z$ ($\forall\, z \in [1,27]$).

$$CD_{j[1,15]} \begin{pmatrix} E_1 VT_1 & \cdots & E_1 VT_{27} \\ \vdots & \ddots & \vdots \\ E_{31} VT_1 & \cdots & E_{31} VT_{27} \end{pmatrix} \tag{1}$$

When the six possible comparisons between the different ASFs were dependent and correlated for the same time window, electrode, and frequency range, we indicated it with the value of 1 in the corresponding matrix position, while using 0 to indicate all other matrix positions. The fact of having worked with Boolean matrices allowed us to conduct the automated computational analysis of the power of the signals. Thus, 1 indicates that the neuronal signal in the frequency domain reflects a reaction to the cut event and 0 when the signal does not respond to the cut event. The fact of having worked with Boolean matrices allowed us to conduct the automated computational analysis of the power of the signals.

Having detected which temporal instants show behaviour related to the event of the cut for each electrode in each frequency band, we could then study how these results were distributed along the timeline and detect whether there were time intervals with higher concentrations of results. To observe the temporal distribution of the neuronal responses to the cut event, we calculated the time between the same time windows in the different electrodes and frequency ranges (Eq 2).

$$\textit{Temporal distribution of reactions due to the cut}_{1x27} = \sum_{k=1}^{31} \sum_{j=1}^{15} CD_j\left(E_k VT_z\right) \tag{2}$$

To do this, we calculated the sum of the matrices *CD* referring to the 15 frequency bands and then the sum of the columns of the resulting matrix. In this way, we obtained a vector of 27 positions, each referring to a time window, which gives us an overview of how the neuronal reactions due to the cut event were distributed over time. Each value of the resulting vector represented the number of sliding windows that cover the same time range ($VT_z$) where a neuronal response was detected as a consequence of the cut in all electrodes ($E_k \; \forall \; k \in [1,31]$) and all frequency bands ($j \; \forall \; j \in [1,15]$).

To locate those neuronal responses from the results detected as a consequence of the shot change by cut that constituted a representative variation in the ERD/ERS, we applied a slope analysis in each time window of the signal in the frequency domain indicated by the CDj[1,15] matrix as dependent and correlated between the 4 ASF's. We set as a condition that the time intervals detected in a particular electrode and frequency range must have a representative and monotonous inclination with the same sign in all four ASFs. Then we located those results that represented a significant neuronal inhibition or excitation due to the shot change by cut.

To apply the slope analysis procedure, we needed to know the threshold values from which we could define a slope as an indicator of a representative variation in the power change. We do not apply the relative change between time points because a short oscillation in the signal can distort the results obtained. We are interested in detecting ascending or descending progressions over various time samples. For this reason, it is more appropriate to locate monotonous slopes that present a representative inclination.

To establish the value that represents a significant increase or decrease in the power change we defined the threshold value using the slopes from the time windows comprised by the samples before the shot change by cut as a reference. As we needed to study the slope in the time windows with the assurance that these slopes were monotonous, we needed to determine the thresholds for the vectors of six and three contiguous samples in the timeline. We would confirm the occurrence of a monotonous slope if inside the six samples comprising a time window we had a three-sample slope with the same signal and a representative inclination [32]. This baseline process was applied to 20,460 slope values for time windows of six samples and 26,040 slope values for time windows of three samples. From this process, we defined the threshold value for sliding windows of six samples at ±19,95° and for sliding windows of three samples to check for monotony at ±25,43°. Therefore, we considered a slope as representative when in the evolution of power change the angle of increase with respect to 0° was greater than the positive value of the threshold or less than the negative value.

## ERD/ERS study of the signals identified as depending on the cut

Once the neuronal response triggered by the shot change by cut was located, we then analyzed its ERD/ERS in the recorded signal. Through an ERD/ERS analysis, we can identify and quantify the percentage of variation in the activation (synchronization) or inhibition (desynchronization) of neurons recorded by an electrode for a frequency band in each film. This representation in relative terms of the power change is useful to identify the neuronal behaviour related to the cognitive and motor system in the results obtained [36]. To calculate the ERD/ERS for each electrode in each frequency band we applied Eq 3 [37, 38].

$$ERD = \frac{Baseline - Test}{Baseline} X \; 100 \tag{3}$$

To quantify the variation of the power change that occurs in an electrode in a certain frequency band, we decided to use a baseline from the values of the power change prior to the instant of the cut, in order to obtain the ERD/ERS. We took this baseline as a reference

because, as we were interested in analyzing the connection that occurs between two shots joined by a cut, it was more useful to analyze the variation due to the shot change in the film's temporal continuum than comparing it with a state of neutral rest.

## Results

The results obtained based on the proposed methodology allow us to locate and determine the electrodes, time window and frequency range of the neuronal activity triggered by the cut event, and also to identify which of these reactions are the most representative in terms of neuronal rhythm variations. In order to detect the time windows and brain areas where we can find neuronal activity associated with the shot change in the first second after the cut, we analyzed the distribution of those responses located as correlated and dependent throughout the set of the six pairs of comparisons between the four ASFs in each electrode and frequency band over the time sample. Based on the methodology described, we analyzed the power in each of the four ASFs through 12,555 time windows, reviewing 27 time windows for each one of the 31 electrodes in 15 frequency ranges. Through this analysis we found that 128 time windows have a correlation and dependence structure for all four ASFs (the same 128 time windows for each ASF), and in 33 of these time windows the power change evolution shows a slope with an inclination above the average in all four ASFs (the same 33 time windows for each of the four ASFs). The positive responses identified have a temporal distribution concentrated in certain time windows, electrodes and frequency bands. The neuronal responses appear after the cut for specific frequency bands and specific electrodes in specific time windows, as will be discussed in this section.

As described in the methodology section, we analyzed the distribution of all the time windows in which neuronal activity dependent on the cut in the four ASFs was detected through the permutation test and the Spearman correlation. At this intermediate point in the application of the methodology, before the study of the slopes in the power change samples, we can already identify the time windows where there is a relationship between the cut event and the behaviour of a specific neuronal area in a specific frequency band. We found a dependence relationship and correlation in the same time windows for the same electrode and frequency band in 128 time windows, that is, 1.02% of the sliding windows analyzed, of which 88.25% contain power samples after the cut event. The results detected as positive before the cut event may be due to the average duration of the *Bonnie and Clyde* and the *Whiplash* shots, being 1235 ms and 1520 ms, respectively. This average duration can have an influence on the power samples of the ASFs for these two films around 1000 ms before the cut due to the previous cut event. Specifically, out of the 15 positive results obtained in the analysis of dependence relations and correlations whose time windows do not have power samples after the cut, six are located in the first time window containing samples from -1000 ms to -688 ms. When applying the slope analysis of these time windows before the cutting event, the only positive result obtained without power samples after the cut is also in that same time window.

The distribution of the positive results along the 6-sample time windows is represented in Fig 2. This figure allows us to visualize the time windows with a dependence and correlation in the neuronal response between the four ASFs, as it shows the number of electrodes per frequency band with responses related to the cut event. Fig 2 represents the reactions detected throughout the two seconds studied (one before and one after the cut) to verify that the detections are concentrated after the cut event, and thus confirm the validity of the methodological process. Through this procedure, we can verify that there is actually a dependence and correlation between the cut event and the neuronal signal recorded, and we are able to identify the time windows in which this dependence occurs. Because the sliding windows contain time

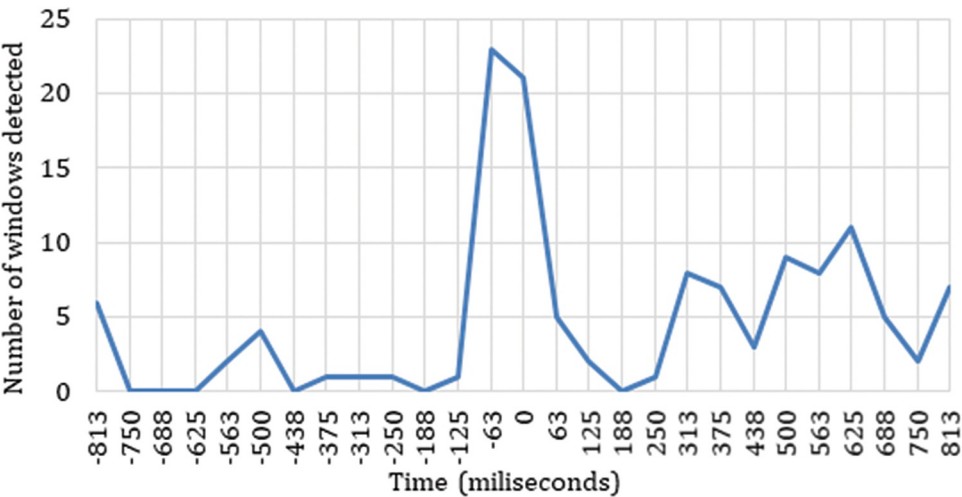

**Fig 2. Distribution of the sliding windows detected as neuronal responses triggered by the cut event.**

windows of six consecutive power samples, the results appear as positive in the two windows before the time window centred on the cut event (0 ms). This is because these time windows show results influenced by the power samples after the cut event. The time window marked as 0 ms corresponds to time window number 14, consisting of the three samples just before the cut and the three samples just after.

Fig 2 shows how a large concentration of neuronal responses related to the cut event are accumulated around the shot change. The time windows centred in t = -63 ms and t = 0 ms include positive correlations, due to the use of sliding windows, which in these cases contain power samples after the cut event. In the time window centred at -125 ms there is only one positive data point, while in the window centred at -188 ms all the results indicate that there is no relationship between the cut and the neuronal response. The time window centred at -125 ms contains one sample after the cut event, while the window centred at -188 does not contain samples after the cut. Taking into account that each time window contains six samples, and that the time window with the highest number of positive results is the one centred at -63 ms followed by the one centred at 0 ms, and that there is a large decrease for the one centred at 63 ms, we can infer a strong dependence relationship between the neuronal responses and the cut event in the first two samples after the cut, that is, between 0 and 125 ms. This result implies an effective detection of the cut event through the data recorded in the electroencephalogram.

In addition to this accumulation zone around the cut event, Fig 2 shows two other time intervals with significant activity triggered by the shot change. One of them occurs between 250 ms and 438 ms and the other between 438 ms and 750 ms. Finally, one last growth trend is visible from 750 ms through to the end of the sample.

The greater number of electrodes recording neuronal responses caused by the shot change due to the cut are around t = 0. We observe that it occurs mainly in the frequency bands related to theta; specifically, they appear with great intensity in high theta, as shown in Table 1.

Once the electrodes that produce neuronal responses to the cut event in particular frequency bands have been detected, it is interesting to analyze their ERD/ERS to classify the neuronal response occurring. Figs 3 and 4 show the ERD/ERS of the four ASFs for the high theta frequency band in two electrodes (Oz and Pz) that register neuronal responses to the cut event in the time windows indicated. The period between 0 and 313 ms is highlighted because this is the period after the cut, covered by the results indicated in Table 1.

**Table 1. Number of correlations and dependencies detected for sliding windows between -63 ms and 63 ms in Theta frequency bands.**

|  | -63 ms. | 0 ms. | 63 ms. |
|---|---|---|---|
| Theta | 6 | 5 | 1 |
| Low theta | 3 | 3 | 0 |
| High theta | 13 | 11 | 4 |

Figs 3 and 4 show how the Oz and Pz electrodes recorded a neuronal response to the cut event in the period between 0 ms and 313 ms in high theta frequencies, reflecting a synchronization in neuronal rhythms. The same synchronization occurs in low theta, high theta, and theta frequency ranges on all the electrodes identified as related to the cut event in this same time interval. Neuronal responses to the cut event in the first 250 ms after the shot change in the theta band are shown in electrodes P7 (theta, low theta), P5 (theta, low theta), P3 (theta, low theta), Pz (theta, high theta), and P6 (theta) located in the parietal cortex. In this same time interval for the high theta frequency range, positive results are also located in the frontal lobe (Fz and F4), the frontocentral (Fc1 and Fc2), and the occipital (O1 and Oz). Specifically, in the theta and low theta frequency bands a left lateralization was detected in the electrodes showing synchronization triggered by the cut event during this time interval.

Our analysis of the results also revealed neuronal responses to the cut event in the delta frequency band. The timing of the neuronal responses to the cut event in delta and theta exhibit clear differences in their distribution, as can be seen in Figs 5 and 6. There are two trends in the signal, one for the frequency ranges covered by delta (Fig 6) and another for the frequency ranges covered by theta (Fig 5).

Analyzing how the results detected as responses to the cut event are distributed across the time windows for the different electrodes, we can locate accumulations of positive results in certain frequencies and time windows. We detected reactions to the cut event in the theta frequency band during the first 125 ms after the cut, between 254 ms and 438 ms, and at the end of the sample starting at 688 ms. Our analysis of ERD/ERS neuronal rhythms indicates that the

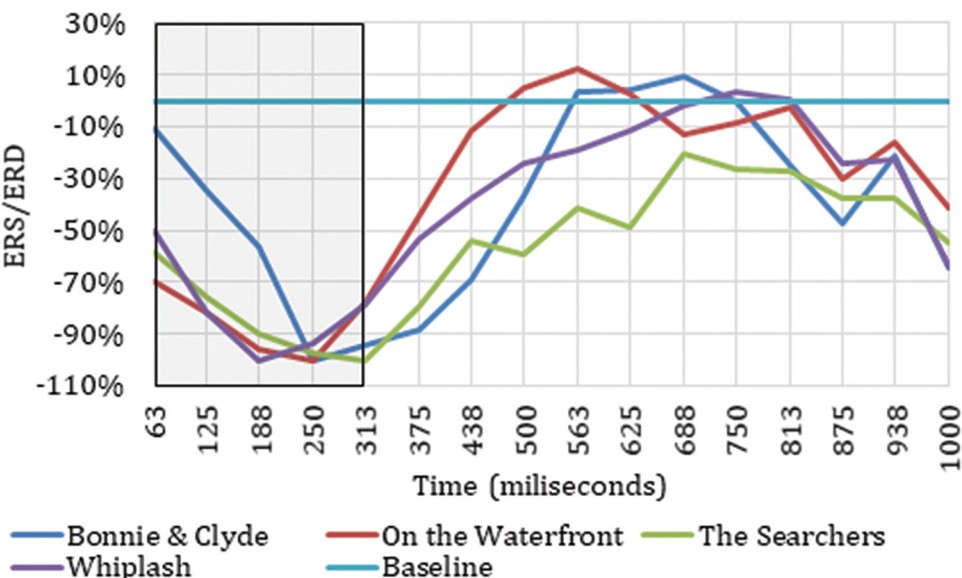

**Fig 3. ERD/ERS from the four film excerpts for Pz electrode in high theta.**

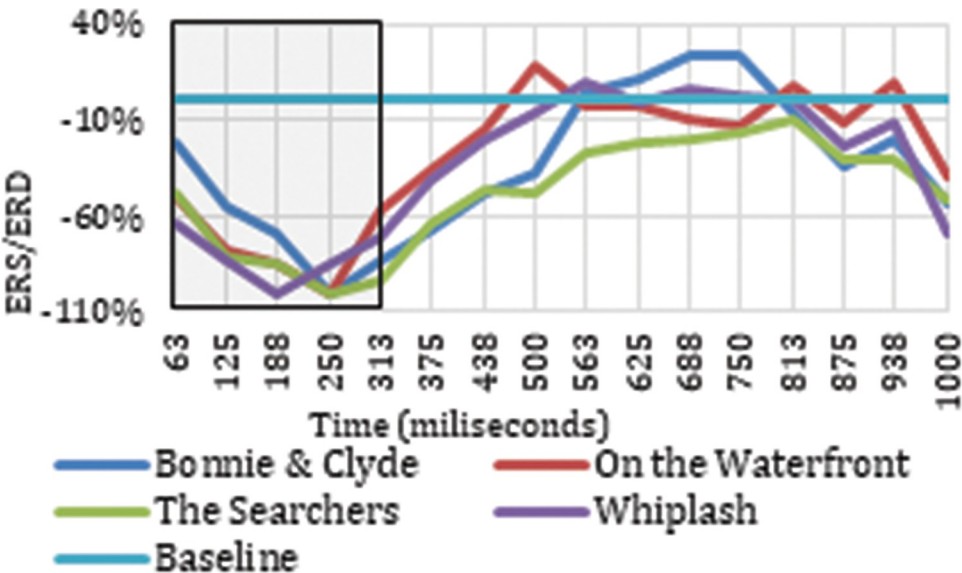

**Fig 4. ERD/ERS from the four film excerpts for Oz electrode in high theta.**

neuronal response detected in the first moments after the cut and at the end of the sample are synchronizations in the neuronal rhythms, while in the period between 254 ms and 438 ms a desynchronization appears. On the other hand, the neuronal responses to the cut event in the delta frequencies appear between 500 ms and 688 ms, especially in low delta, and an analysis of their ERD/ERS reveals a desynchronization in neuronal rhythms.

If we study the spatial distribution of the neuronal responses to the cut according to their corresponding electrodes, for theta frequencies between 250 ms and 438 ms the correlations in theta occur mainly in the right hemisphere (F8, Fc2, C6, P6, Pz) and in the occipital area (O1

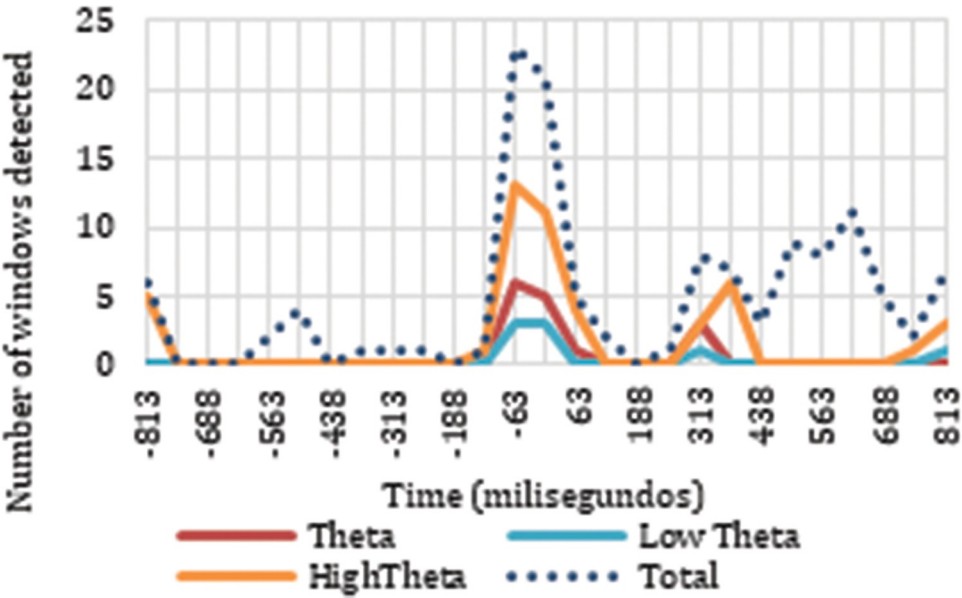

**Fig 5. Distribution of sliding windows detected as triggered by the cut in theta frequencies.**

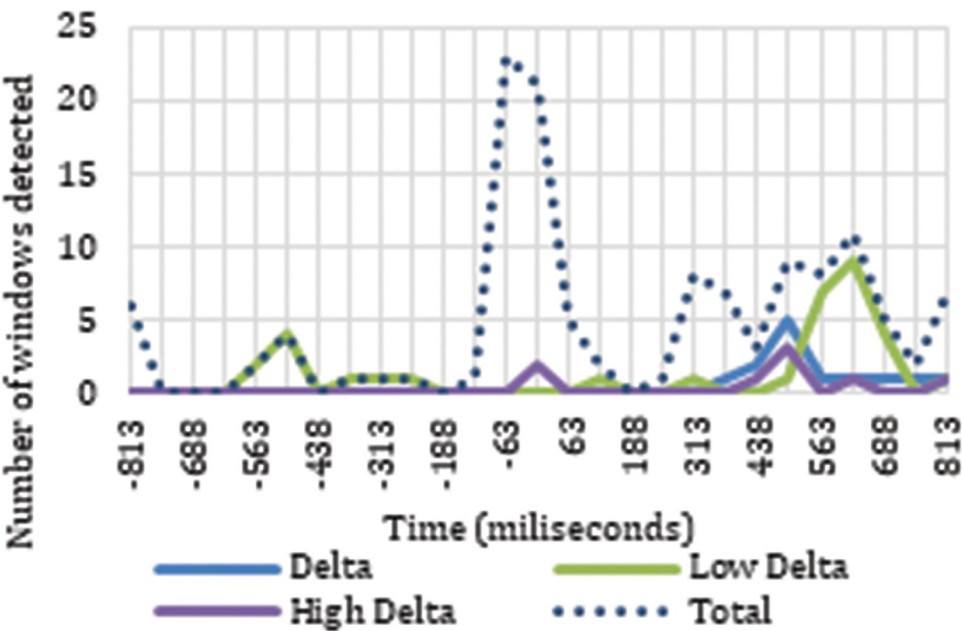

**Fig 6. Distribution of sliding windows detected as triggered by the cut in delta frequencies.**

and Oz) with a desynchronization tendency. At the end of the sample (688 ms—1000 ms), the neuronal reactions dependent on the cut appear in the theta frequency ranges with a tendency toward synchronization; specifically, for low theta in the right frontopolar area (FP2), and for high theta in both hemispheres of the frontal area (F4 and F8) and in the left temporal area (T7). The results determined as dependent on the cut event between 500 ms and 688 ms in the delta frequency bands are concentrated in the parietal area (P5, P3, Pz, P4). Specifically for the low delta band, the middle part of the cortex is shown in reaction to the cut, crossing the frontal (F3 and Fz), frontocentral (Fc1 and Fc2), central (C5, C3, Cz, and C4), and parietal area (P5, Pz, P4, and P6).

The next step according to the methodology described is to extract from the results those in which there occurs significant variations in the power of all the time windows detected as dependent on and correlated to the input of the shot change. To ascertain this, we studied its slopes, locating and identifying significant synchronization or desynchronization processes in the neuronal rhythms (ERD/ERS) produced after the shot change by cut.

The final result identifies the time windows, electrodes, and frequency bands where there are a significant variation in the power change as a reaction to the input of the shot change by cut. We obtain this result by combining the permutation test processes, the Spearman correlation, and the detection of slopes with the mean defined as a threshold. Out of all the 12,555 ($27Vt$ X $15Bf$ X $31E$) possible time windows when we transformed the EEG to the frequency domain, we identified 33 in which, in any of the 31 electrodes in any of the 15 established frequency ranges, a representative variation is detected showing dependence and correlation structures related to the power of the four ASFs in the two seconds analyzed. Of these 33 localized cases, 32 are in time windows that contain samples after the cut event, indicating that they have a clear relationship with the shot change. Fig 7 shows the distribution of the time windows identified that have dependence and correlation among all four ASFs and at the same time have representative slopes. Fig 7 also compares these results with the distribution of the

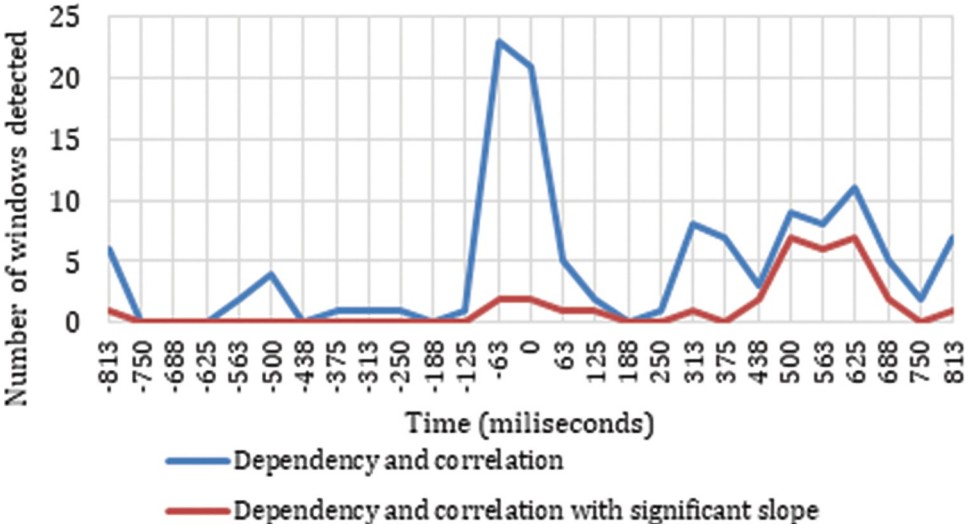

**Fig 7. Sliding time window distribution detected as significant variations in power change in response to the cut event.**

windows that only have dependence and correlation structures among all four ASFs at the same time.

Fig 7 shows how the accumulation of time windows that show significant variations in the power change in different electrodes for different frequency bands occurs specifically in two time intervals: one located between -63 ms and 63 ms, and the other between 375 ms and 750 ms. In the data analyzed (two seconds around the cut), we identified that 0.02% of the time windows studied represent a characteristic neuronal response directly related to the shot change event. Comparing these results with the time windows identified as having only dependences and correlations with the cut event (Fig 7), we find that it is in the interval between 375 ms and 750 ms where the results are more similar, while there is a big difference between -63 ms and 63 ms. Fig 8 shows the representative neuronal responses detected in the power change

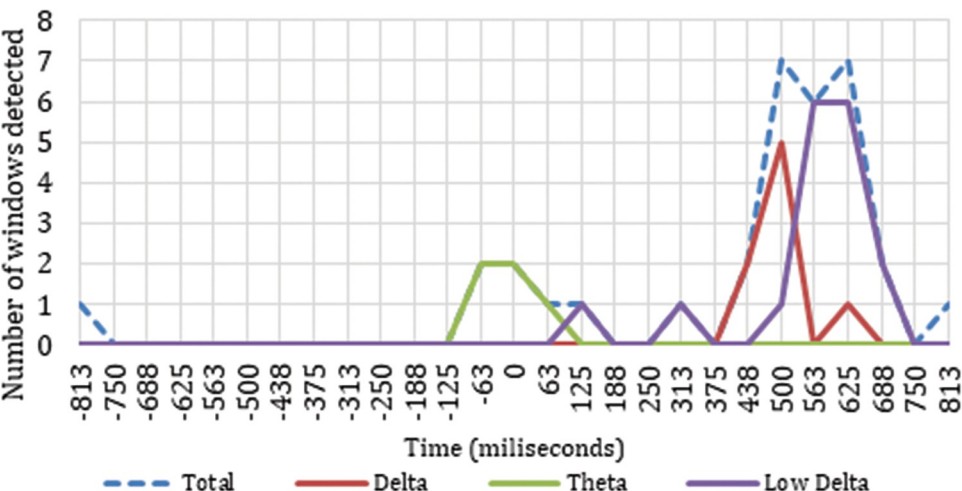

**Fig 8. Distribution of time windows that have variations with monotonous and significant slopes in the power change as a response to the cut event.**

due to the cut event for all the electrodes, distributed along the timeline and broken down into the most representative frequency ranges detected.

Fig 8 shows a hard concentration of significant results in delta frequencies between 375 ms and 750 ms. This time period in delta frequency has an increase starting at 375 ms and ending at 563 ms, with a maximum at 500 ms. Low delta frequency starts to increase at 438 ms and ends at 750 ms, with a maximum between 563 ms and 625 ms. Theta frequency band accumulates the positive time windows at the moment of the shot change, specifically between -63 ms and 63 ms.

Figs 9, 10 and 14 show the placement of the electrodes in which representative reactions were detected at each time period and frequency band as expressed in Fig 8. In each figure showing the distribution of the electrodes, the number of time sliding windows for the same electrode in the same frequency band where positive results are obtained is represented in three colours: red for three positive time windows (the maximum), yellow for two positive time windows, and green for one positive time window (the minimum). Once the electrodes that are reacting significantly to the cut event for a certain frequency band have been detected, the ERD/ERS analysis tells us the type of variation they represent in terms of neuronal rhythms, as shown in Figs 11–13.

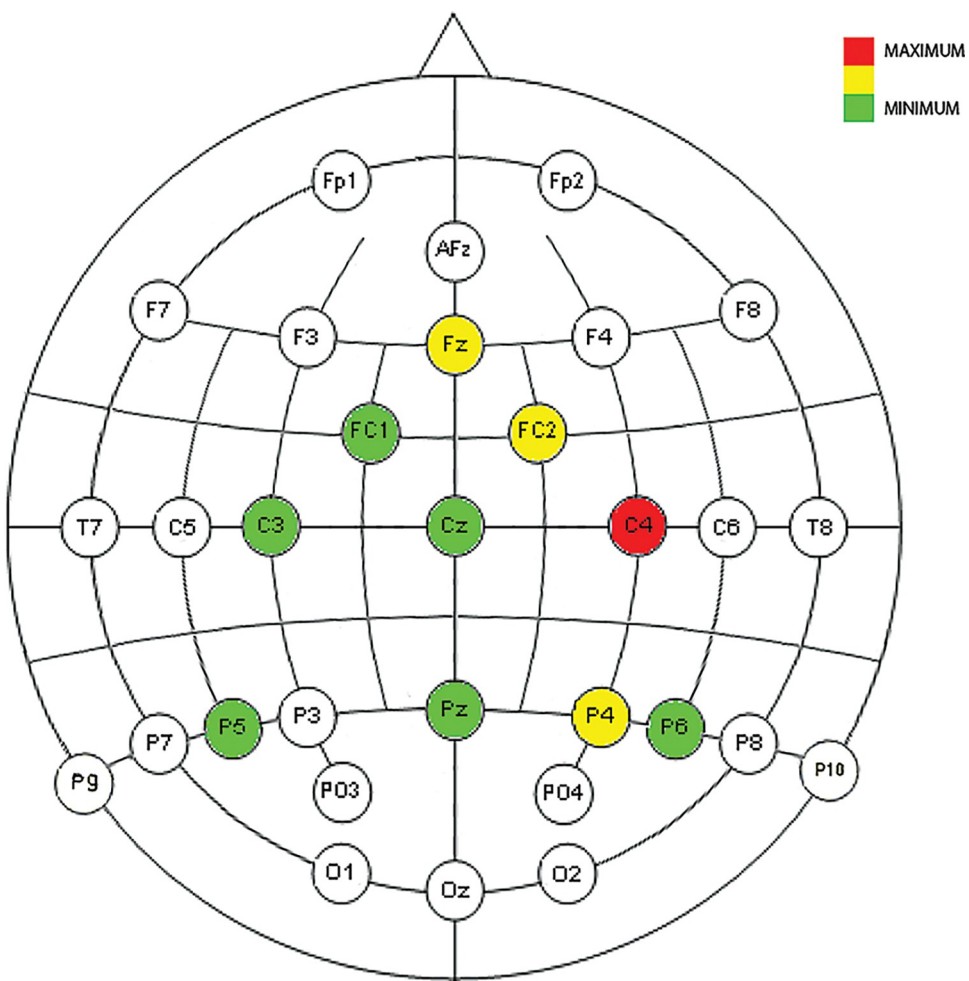

**Fig 9. Neuronal distribution map of the electrodes that detect correlated, dependent, and significant monotonous slopes in the power change for all four ASFs in low delta between 438 ms and 750 ms.**

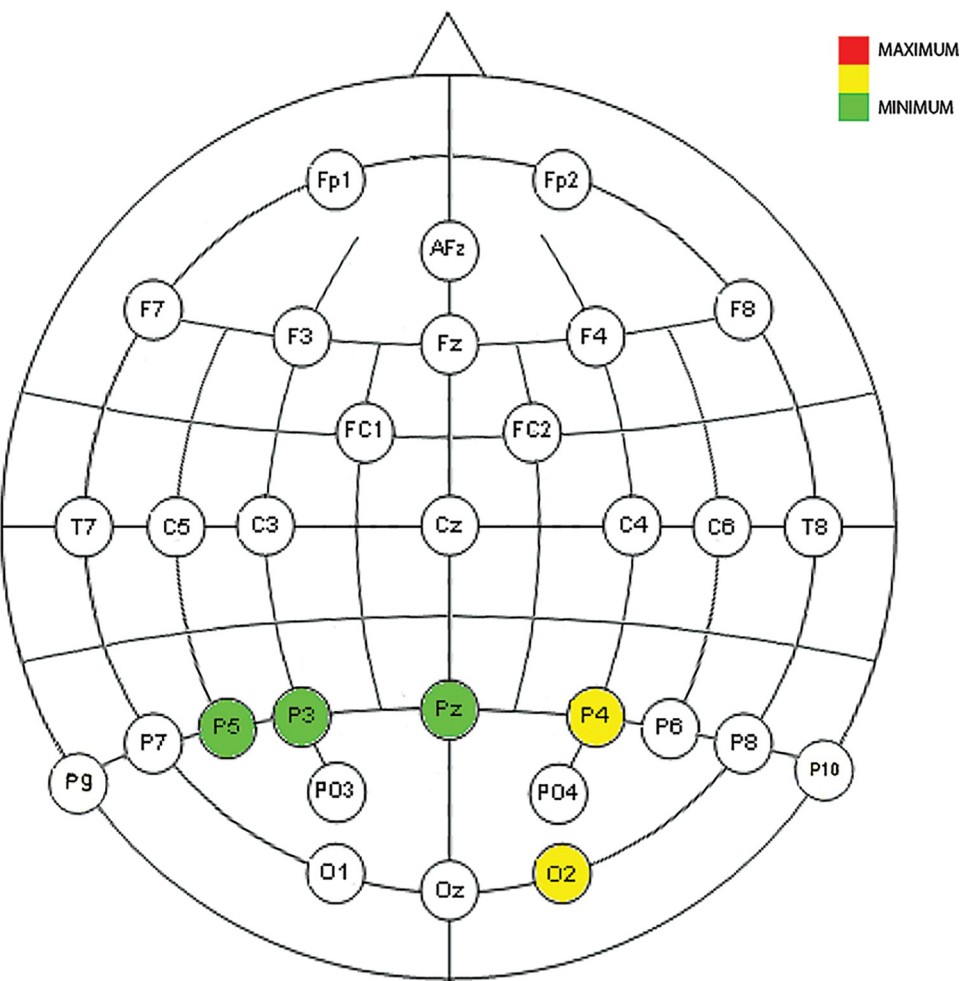

**Fig 10. Neuronal distribution map of the electrodes that detect correlated, dependent, and significant monotonous slopes in the power change for all four ASFs in delta between 375 ms and 563 ms.**

As reflected in Figs 11 and 12, in the time between 375 ms and 750 ms, there is a tendency towards desynchronization for the delta and low delta frequency ranges. The electrodes that recorded these results (Figs 9 and 10) were distributed in the middle parietal area: specifically, electrodes P5, Pz and P4 in both frequency ranges, and also P3 in delta and P6 in low delta. If we focus on the cortical distribution of the electrodes in low delta (Fig 9), which is where most of the results are found, the time period where results are located is specifically between 438 ms and 750 ms, and this also extends to the parietal zone, to the middle zone of the frontal (Fz), frontocentral (Fc1 and Fc2), and central (C3, Cz and C4) areas. The results located in theta are found in the time windows between -63 ms and 63 ms, which means time windows with power change samples up to the first 250 ms. In this period, a tendency towards synchronization appears in the ERD/ERS analysis (Fig 13) and the electrodes (Fig 14) are located in the parietal area (Pz and P6).

Thanks to the methodology applied, we have not only been able to locate the electrodes, time periods, and frequency bands where there is a correlation among all ASFs due to the event of the shot change by cut. We have also managed to identify those results in which the most significant variations in the power change reveal to us how neuronal rhythms vary when

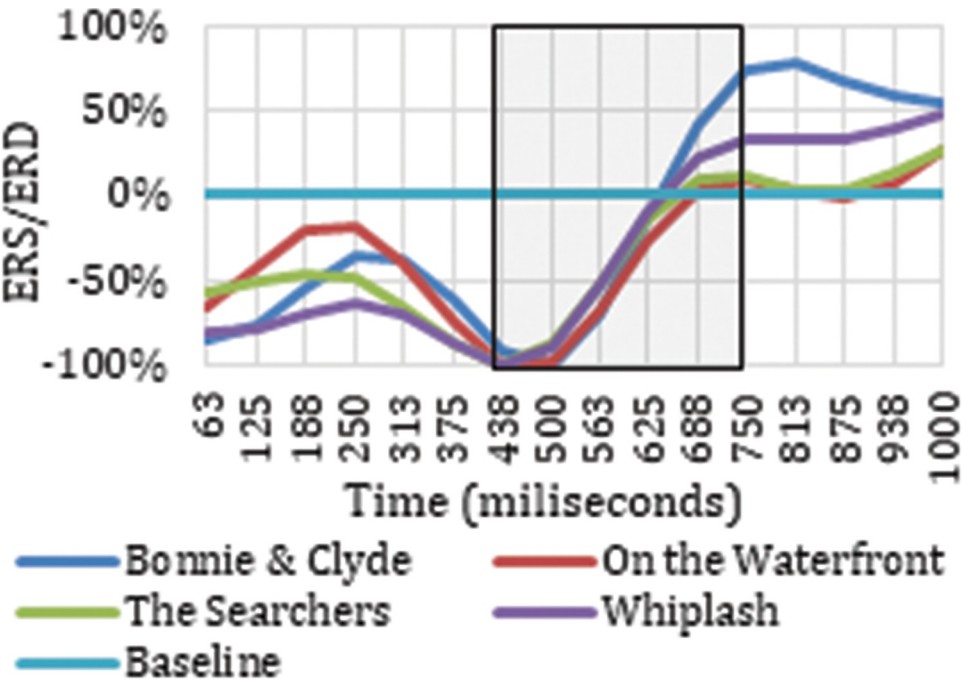

**Fig 11. ERD/ERS from the four film excerpts for P4 electrode in low delta.**

the viewer's brain processes the shot change by cut. The results obtained by analyzing the variations, correlations, and dependences in the power change of the ERD/ERS indicate clear neuronal behaviour that we can relate to the cut event of as patterns of reaction to that event.

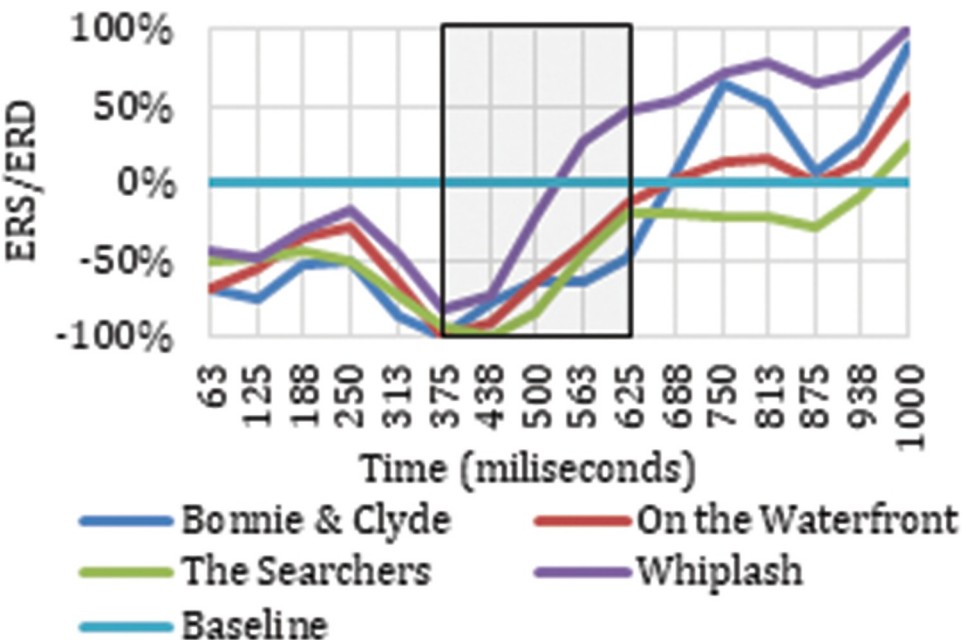

**Fig 12. ERD/ERS from the four film excerpts for Pz electrode in delta.**

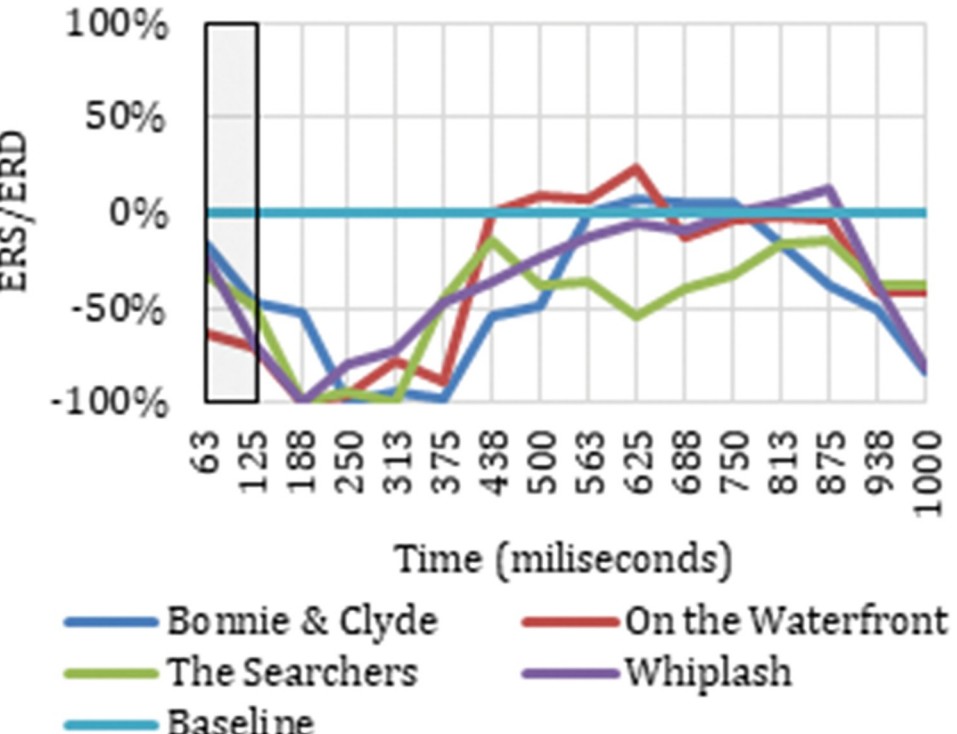

**Fig 13. ERD/ERS from all four film excerpts for Pz electrode in theta.**

## Discussion

The first aspect worth highlighting in our analysis of neuronal patterns in reactions to the cut event is that the proposed methodology clearly reveals dependent and correlated behaviour among the four ASFs due to the shot change, as can be observed in Fig 2. The evidence shows that there is a clear pattern of neuronal responses to the shot change by cut. This pattern is triggered immediately at the instant of the cut in terms of theta frequency during the first 188 ms and is reflected in the intensity of the variation of neuronal rhythms (ERD/ERS) between 438 ms and 750 ms in the low delta frequency. The brain area most reactive to the cut, in all frequency ranges, appears to be the parietal area.

The neuronal response pattern that produces a more representative variation in neuronal rhythms occurs in the delta frequency range between 375 ms and 750 ms, especially in low delta. The results obtained in low delta and delta tend towards desynchronization of neuronal rhythms. The activity detected in delta is located in the middle zone of the parietal, frontal, frontocentral and central areas, without reflecting lateralization for all. Desynchronization of delta rhythms is considered related to cognitive processes, reflecting the suspension of the default mode network, or DMN [39], which is associated with the beginning of conscious activity. In this sense, Li, Mai and Liu [40] associated the DMN with emotional perception and empathy in social relationships.

The most immediate neuronal response pattern triggered by the cut event that we have identified occurs in the theta frequency range, especially in high theta. This dependence detected in the theta band reflects ERS activity during the first 188 ms after the cut. The main neuronal zone where we detected this response pattern was the parietal area, while in the case of high theta it also extended to the central, occipital, and middle areas of the frontal and frontocentral lobes. In this first period, we detected a tendency towards a left lateralization in

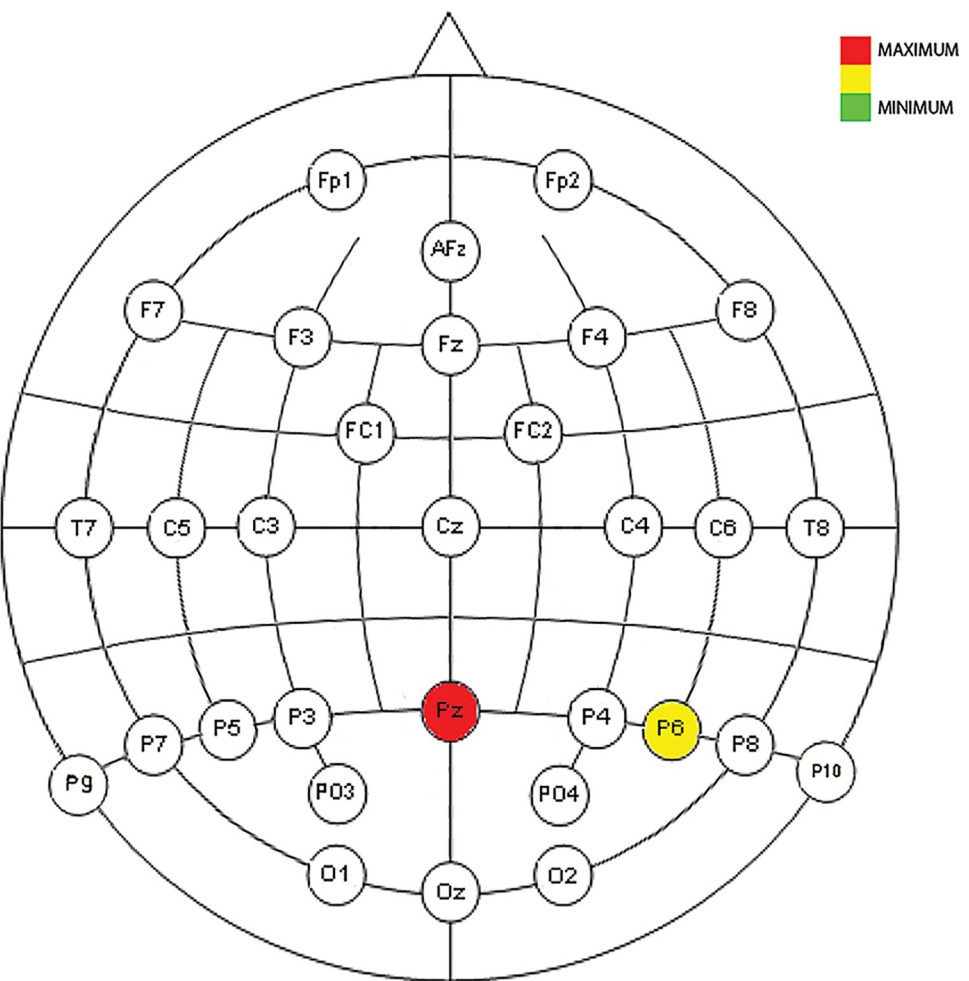

**Fig 14. Neuronal distribution map of the electrodes that detect correlated, dependent, and significant monotonous slopes in the power change for all four ASFs in theta between -63 ms and 63 ms.**

neuronal responses, something typically related to cognitive processes of language [41]. Synchronization in the theta frequency band during the first 250 ms is also associated with unconscious emotional processing of the stimulus [42].

Synchronization in the theta band is considered mainly a reflection of activity in the hippocampus, and it is related to the encoding of new information and processes related to memory [43–45]. The activity in the frontal area when memorizing or remembering is mainly reflected in theta frequencies [46]. In this sense, the results show a pattern of neuronal responses consistent with the results described by the cited authors, which means taking into consideration the possibility that a reflection of the activity of the hippocampus is being detected in the neuronal records detected because of the cut.

In addition, there are two previous studies that relate the cognitive processing of the cinematographic shot change by cut with neuronal activity located in the hippocampus, one though magnetic resonance [12] and the other though electroencephalogram [47]. In relation to the research by Ben-Yakov and Henson [12], in their conclusions they affirm that the participation of the hippocampus in the perception of cinematographic edition allows a sensorial

continuous experience to be formed from the representation of discrete events registered though memory.

Various studies hold the hippocampus responsible for time perception and associate it with memory processes [48–50] through so-called *time cells* [51]. Along the same theoretical lines, it has been suggested that the hippocampus encodes time through memory [52]. These conclusions support the hypothesis that filmic time is generated through the cut [53], and the proposal that editing is responsible for the articulation of the discourse of the film narrative. Films have a specific time duration that does not usually correspond to the length of the filmic time narrated. The film generates its own temporal structure in the development of its narration. Filmic time is a perception that must be encoded [53], as Teki, Gu and Meck [52] suggest, with the perception of our surroundings through memory processes, and the hippocampus is responsible for encoding reality in temporal terms.

Researchers have also linked the hippocampus to the encoding of our perception of space in a similar way to how it encodes our perception of time [54–56]. According to these theories, the hippocampus fulfils the function of space-time encoder, and thus supports the spectator's need to encode filmic space and time, as defined by Burch [21]. Burch defines the film as an abstraction of two *découpages*; the temporal *découpage* and the spatial *découpage*. For Burch, a film is a succession of shots, where each shot is a fragment of space and time, and through the codification of these fragments of space and time the filmic discourse is articulated, generating filmic space and time. The results obtained in this study indicate the cut as a trigger for synchronizations in theta, which can be associated with hippocampal memory processes, suggesting that temporal and spatial perception of a film could be codified through the cut in a way similar how such perception is processed in reality, as the theories of cinematographic cognitive ecologism affirm [2]. The results obtained seem to support the idea that the shot change by cut may be responsible for the generation of our sense of filmic time and space.

## Conclusions

By means of a novel methodological approach, we have been able to detect neuronal modulation patterns triggered by shot changes by cuts. The experiments though electroencephalogram that analyse the shot changes by cut that we can find in the previous literature are carried out using audiovisual material filmed for the experiment, since the fact of working with film excerpts is a problem due to the large number of inputs that surrounded the cut event. Our methodological approach faces this limitation and, based on film excerpts, achieve isolate those neuronal reactions consequence of processing the shot change.

The neuronal modulation patterns triggered by shot changes by cuts detected are located in theta and delta frequency bands. Specifically, in the theta band, there are two time periods in the second after the cut where ERS activity has been identified in several electrodes, and a third time period where ERD is located. The electrodes that registered synchronization in the theta frequency band did so in the first 188 ms and after 750 ms through to the end of the sample. The desynchronization detected in theta appeared between 250 ms and 750 ms. On the delta frequency ranges, electrodes detecting synchronization did so between 375 ms and 750 ms, especially in low delta. The cortical area where more electrodes detected neuronal responses due to the cut event for theta and delta frequencies is the parietal area.

The modulation in neuronal rhythms detected in delta, possibly related to the default mode network, together with the modulation detected in theta, possibly related to processes of encoding and memory access, seem to indicate that the spectator's cognitive system needs to assimilate the shot change through a neural process associated with message encoding and decoding, especially in light of the detection of power change modulations in the left

hemisphere in theta frequency. The relationship between the results obtained and the activity of the hippocampus is considered of special interest due to the neuroscientific theories associating the hippocampus with our space-time encoding of reality.

We can conclude that in an analysis from the perspective of the spectator's cognitive mechanisms we can detect electroencephalogram patterns in the neuronal responses of spectators exposed to shot changes introduced by continuity editing cuts. These patterns refer to a hippocampal activation and reflect a DMN suspension. The theorical relation between hippocampal and codification, specifically in time and space codification reveals an interesting point to understand the nature of the shot change by cut. To be able to affirm this relation with total certainty, further experiments are needed to examine this possibility of space-time articulation in more detail.

## Author Contributions

**Conceptualization:** Javier Sanz-Aznar, Lydia Sánchez-Gómez, Carlos Aguilar-Paredes.

**Investigation:** Javier Sanz-Aznar, Luis Emilio Bruni.

**Methodology:** Javier Sanz-Aznar, Carlos Aguilar-Paredes.

**Resources:** Luis Emilio Bruni.

**Software:** Andreas Wulff-Abramsson.

**Supervision:** Lydia Sánchez-Gómez, Luis Emilio Bruni, Carlos Aguilar-Paredes.

**Validation:** Lydia Sánchez-Gómez, Carlos Aguilar-Paredes.

**Visualization:** Andreas Wulff-Abramsson.

**Writing – original draft:** Javier Sanz-Aznar.

**Writing – review & editing:** Lydia Sánchez-Gómez, Luis Emilio Bruni, Carlos Aguilar-Paredes.

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
