## [Decision Letter · Decision Letter 0]

4 Aug 2021

PONE-D-21-20109

Neural Responses to Shot Changes by Cut in Cinematographic Editing: An EEG (ERD/ERS) Study

PLOS ONE

Dear Dr. Sanz-Aznar,

Thank you for submitting your manuscript to PLOS ONE. After careful consideration, we feel that it has merit but does not fully meet PLOS ONE’s publication criteria as it currently stands. Therefore, we invite you to submit a revised version of the manuscript that addresses the points raised during the review process.

We look forward to receiving your revised manuscript.

Kind regards,

Dragan Hrnčić

Academic Editor

PLOS ONE

Journal Requirements:

Reviewers' comments:

Reviewer's Responses to Questions

**Comments to the Author**

1. Is the manuscript technically sound, and do the data support the conclusions?

Reviewer #1: Partly

2. Has the statistical analysis been performed appropriately and rigorously? 

Reviewer #1: I Don't Know

3. Have the authors made all data underlying the findings in their manuscript fully available?

Reviewer #1: Yes

4. Is the manuscript presented in an intelligible fashion and written in standard English?

Reviewer #1: Yes

5. Review Comments to the Author

Reviewer #1: The manuscript is concerned with detecting cuts separating film scenes from the scalp EEG recorded from subjects watching films. The authors identify EEG electrodes and frequencies that seem suitable for this purpose. The study and its findings are appropriate for this journal. However, I do have some concerns regarding the methodology that should be resolved before I can recommend the manuscript for publication.

Major comments / questions:

1. My main concern is that a total of 12555 hypotheses were tested statistically, yet there is no mention of how the number of false positives were controlled, if at all. Some form of control of false positives should be used and mentioned (e.g. Bonferroni, Benjamini-Hochberg); otherwise it should be carefully and convincingly argued why no correction of p-values was performed.

2. How was temporal precision achieved given that at delta / theta frequencies the oscillation period T = 1/f is (much) larger than the temporal resolution?

3. Why was frequency analysis used and not (standard) ERP analysis? Did the latter fail to locate cuts between shots?

4. EEG is not capable of detecting signals from the hippocampus, and cortical theta may have distinct characteristics from hippocampal theta (see e.g. ‘Cortical theta wanes for language’, D. Hermes et al., Neuroimage (2014), 85: 738-748). This study does not elucidate any role of the hippocampus, yet it may be speculated that the hippocampus could be involved in sequential processing based on previous studies. Therefore, I recommend to remove the last sentence from the abstract, and to refocus the discussion on the results of this work.

Minor:

5. p.18 of the compiled pdf: It is quite uncommon to use the slope (in degrees) to measure signal changes. Why wasn’t the relative increase/decrease between time points used?

6. p.17: equations should be moved to where they are first mentioned.

6. PLOS authors have the option to publish the peer review history of their article (what does this mean?). If published, this will include your full peer review and any attached files.

Reviewer #1: No

---

## [Author Response · Author response to Decision Letter 0]

28 Aug 2021

Journal Requirements:

Done

The data could be available by https://osf.io/exbt7/.

P.27: We have added this link in the manuscript in a section called Open Research badges

P.7: We have moved the ethics statement to the Methods section

4. While revising your submission, please upload your figure files to the Preflight Analysis and Conversion Engine (PACE) digital diagnostic tool, https://pacev2.apexcovantage.com/. PACE helps ensure that figures meet PLOS requirements.

Done. We’ve uploaded the figure files to PACE

Reviewer's Comments:

2. Has the statistical analysis been performed appropriately and rigorously? 

Reviewer #1: I Don't Know

We have published a methodological paper in "Lecture Notes in Computer Science", that is a Scopus SJR peer-reviewed indexed journal. We have the methodology validated by a peer-reviewed indexed journal focused on computational statistics.

Manuscript Page 30: We have changed the reference to the methodology paper used in the manuscript for this one.

Reviewer's Requirements:

Major comments / questions:

1. My main concern is that a total of 12555 hypotheses were tested statistically, yet there is no mention of how the number of false positives were controlled, if at all. Some form of control of false positives should be used and mentioned (e.g. Bonferroni, Benjamini-Hochberg); otherwise it should be carefully and convincingly argued why no correction of p-values was performed.

As the reviewer correctly indicates, the control of false positives should be an aspect taken into consideration for the methodological design. Due to the high number of comparisons we made in the data analysis, we appreciate the reviewer's comment when indicating the importance of referring to it in the manuscript.

Although there is some published neuroscientific research where the permutations test. [1, 2] and the Spearman correlation test [3, 4, 5] have been applied separately, we decided to apply both tests together, because it allows a greater robustness of the results [6] by detecting false positives when the results obtained by both tests are compared [7, 8].

The control of false positives or type I errors when applying the Spearman correlation test is carried out by means of the combined application with the permutations test. There are other strategies to control false positives for Spearman, but since computer systems allow it, the permutations test is considered the most effective. Performing the permutations test and the Spearman correlation test in combination is a fairly typical methodological application, which authors such as Kończak [6] propose as a necessary combination in order to obtain robust results. Using the Spearman correlation test we calculate the possibility of a correlation between a pair of compared signals and using the permutations test we compare possible variations in the data of the signals to know if the dependence between two signals is casual or robust. For this reason, the combination between the permutations test and the Spearman correlation test makes it possible to avoid false positives, since we exclude all those results that are not significant in both tests [7]. A case of false positive detected by means of this methodological combination would be, for example, a comparison between signals that is shown as correlated by the Spearman test, but when applying the permutations test it is revealed as non-dependent, that is, a false positive has been detected when applying the Spearman test.

Based on the comment proposed by the reviewer, the following paragraph has been included in the manuscript:

Manuscript Page 10: “Due to the high number of comparisons performed, it is necessary to avoid the possibility of false positives. The combined use of the permutations test and the Spearman correlation test allow obtaining robust results [6], avoiding type I errors or false positives [7, 8]. The joint use of both tests allows to eliminate false positives because it allows to discard all those results that reflect correlation but do not show dependency, at the same time that we eliminate all those that, even showing dependency, do not show correlation.”

2. How was temporal precision achieved given that at delta / theta frequencies the oscillation period T = 1/f is (much) larger than the temporal resolution?

It is very common not to take the full oscillation for analysis in the frequency domain. It is usual that throughout the oscillation the number of neurons dedicated to the task varies, so that inside to the same frequency band oscillation it is possible to detect variations in the ERD/ERS. It is easy to find investigations in which the ERD/ERS of the theta frequency band is analyzed in intervals of 50 ms [9, 10, 11], which supposes a temporal precision even shorter than ours (62,5 ms). There are previous research that, as we do, analyze the signal in the frequency domain with a temporal resolution of 62.5 ms [12, 13].

For example, in the EEG, theta oscillation can show a clear draw, but if we amplificated the line that draw the oscillation, we can detect that this line is conformed by little oscillations. These little oscillations depending about the number of neurons involved in the process. Once we have discriminated one frequency range from the signal, we analyse these internal oscillations to determine the quantity of neurons involucrate in the frequency range oscillation. Then we can detect if there is a neural activation or inhibition.

The time periods defined to transform the EEG signal to the frequency domain depend, on the one hand, on the time precision required for the experiment and, on the other, that each section contains an enough number of samples to allow a robust transformation. In our case, we required high temporal precision and we consider that performing frequency domain transformations from 16 samples is a fairly common robust transformation for EEG recordings performed at 256 Hz [14].

On the other hand, it is true that we have not defined in the manuscript how we divide the temporal signal to transform it to the frequency domain, so we add the following paragraph in the manuscript:

Manuscript Page 9: “To transform the recorded EEG signal to the frequency domain, it is necessary to divide the signal into time intervals that collect each interval the waveform to be analyzed. Therefore, we take sets of 16 successive EEG samples for each signal belonging to each electrode as interval where apply the fast Fourier transform. Taking into consideration that the EEG has registered 256 samples per second, we obtain a total of 16 samples for each second in the frequency domain, assuming each frequency sample a time interval of 62.5 milliseconds [14]”.

3. Why was frequency analysis used and not (standard) ERP analysis? Did the latter fail to locate cuts between shots?

This question is important. It would be a problem for our recording EEG data if the ERP analysis failed to locate the slices. There are previous investigations that analyze the shot change by cut through the ERP analysis. For example, through ERP analysis, Francuz and Zabielska-Mendyk [15] are able to differentiate between cuts called unrelated from those called related. Also Heimann and her research team [16] are able to differentiate between normative cuts and those that perform an axis jump. The ERP analysis should not only allow locating the shot change by cut, but it should be possible to differentiate variations between the different types of cut.

However, we focused our attention in detecting neuronal activation or inhibition patterns due to the cut event. In this sense, our research is close to a Heimann’s research focused about analysing though ERD/ERS different kinds of camera movement [17]. We have not applied ERP analysis, but it’s probably that we perform it for a new paper.

We have added an explanation about the reviewer comment in the manuscript:

Manuscript Page 4 and Page5: “In the present investigation, we are interested in locating those neuronal areas that show neuronal activation or inhibition, trying to establish a map of those neuronal areas involved in the processing of the shot change by cut. The analysis in the frequency domain allows us to differentiate, depending on the oscillation of the analyzed wave, the degree of neuronal excitation that is taking place at each moment, allowing us to establish a pattern of neuronal activations and inhibitions in reaction to the shot change by cut.”

4. EEG is not capable of detecting signals from the hippocampus, and cortical theta may have distinct characteristics from hippocampal theta (see e.g. ‘Cortical theta wanes for language’, D. Hermes et al., Neuroimage (2014), 85: 738-748). This study does not elucidate any role of the hippocampus, yet it may be speculated that the hippocampus could be involved in sequential processing based on previous studies. Therefore, I recommend to remove the last sentence from the abstract, and to refocus the discussion on the results of this work.

The reviewer comment is right, and we appreciate the advice.

There are various investigations that indicate the hippocampus as the area of the brain that has the main influence on the neural processes triggered by the shot change by cut [18, 19]. In our research, the results obtained are perfectly consistent with this proposal. Therefore, it is important to highlight this aspect in the discussion and in the conclusions.

Based on the reviewer's feedback we have made the following changes:

We have removed the last sentence from the abstract, as reviewer 2 recommend to us: Manuscript Page 2: Sentence removed

We also refocus the discussion according to the reviewer's advice Manuscript Page 24 and Page25:”Synchronization in the theta band is considered mainly a reflection of activity in the hippocampus, and it is related to the encoding of new information and processes related to memory [20, 21, 22]. The activity in the frontal area when memorizing or remembering is mainly reflected in theta frequencies [23]. In this sense, the results show a pattern of neuronal responses consistent with the results described by the cited authors, which means taking into consideration the possibility that a reflection of the activity of the hippocampus is being detected in the neuronal records detected as a consequence of the cut.

 In addition, there are two previous studies that relate the cognitive processing of the cinematographic shot change by cut with neuronal activity located in the hippocampus, one though magnetic resonance to the cut that reflect activity in the hippocampus and match the results obtained in a previous study using magnetic resonance and the other though electroencephalogram [19]. In relation to the research by Ben-Yakov and Henson [18], in their conclusions they affirm that the participation of the hippocampus in the perception of cinematographic edition allows a sensorial continuous experience to be formed from the representation of discrete events registered though memory.”

Minor:

5. p.18 of the compiled pdf: It is quite uncommon to use the slope (in degrees) to measure signal changes. Why wasn’t the relative increase/decrease between time points used?

This is an important question. We considered different options to quantify the power change in the signal and determinate if this change is representative or don’t. We need detecting when the increase or the decrease in the power change keep coherent in successive samples. Because of it, we use the slope analysis searching for monotonous variations.

It’s important justify this methodology. Then, we add a paragraph to the manuscript. 

Manuscript Page 12: “We do not apply the relative change between time points because a short oscillation in the signal can distort the results we obtain. We are interested in detecting ascending or descending progressions over various time samples. For this reason, it is more appropriate to locate monotonous slopes that present a representative inclination”.

6. p.17: equations should be moved to where they are first mentioned.

We are agree. We apply it in two cases.

Manuscript Page 11 and Page12: We have moved the Equation 1 and the Equation 2 following this advice.

Finally we would like to thank the editors and reviewers for their work. The proposed changes indicated have been very successful and have helped to improve our paper. Applying the indicated changes has helped us achieve a more solid and robust manuscript.

We hope that the current state of the manuscript meets the quality requirements that make it worthy of being published in Plos One.

References

[1] E. Maris and R. Oostenveld, “Nonparametric statistical testing of EEG-and MEG-data,” Journal of neuroscience methods, vol. 164, no. 1, pp. 177-190, 2007. 

[2] T. E. Nichols and A. P. Holmes, “Nonparametric permutation tests for functional neuroimaging: a primer with examples,” Human brain mapping, vol. 15, no. 1, pp. 1 - 25, 2002. 

[3] F. Singh, J. Pineda and K. S. Cadenhead, “Association of impaired EEG mu wave suppression, negative symptoms and social functioning in biological motion processing in first episode of psychosis,” Schizophrenia research, vol. 130, no. 1-3, pp. 182-186, 2011. 

[4] A. O. Rossetti, E. Carrera and M. Oddo, “Early EEG correlates of neuronal injury after brain anoxia,” Neurology, vol. 78, no. 11, pp. 796-802, 2012. 

[5] . M. M. Doppelmayr, W. Klimesch, T. Pachinger and B. Ripper, “The functional significance of absolute power with respect to event-related desynchronization,” Brain topography, vol. 11, no. 2, pp. 133-140, 1998. 

[6] G. Kończak, “On testing significance of the multivariate rank correlation coefficient,” Acta Universitatis Lodziensis. Folia Oeconomica, vol. 3, no. 335, pp. 21-34, 2018. 

[7] H. Yu and A. D. Hutson, “A Robust Spearman Correlation Coefficient Permutation Test,” arXiv preprint, p. 2008.01200, 2020. 

[8] J. Stelmach, “Using Permutation Tests in Multiple Correlation Investigation,” Acta Universitatis Lodziensis. Folia Oeconomica, vol. 269, pp. 73-81, 2012. 

[9] C. M. Krause, P.-A. Boman, L. Sillanmäki, T. Varho and I. E. Holopainen, “Brain oscillatory EEG event-related desynchronization (ERD) and-sychronization (ERS) responses during an auditory memory task are altered in children with epilepsy,” Seizure, vol. 17, no. 1, pp. 1-10, 2008. 

[10] Y. Okamoto and S. Nakagawa, “Effects of light wavelength on MEG ERD/ERS during a working memory task,” International Journal of Psychophysiology, vol. 104, pp. 10-16, 2016. 

[11] C. M. Krause, P.-A. Salminen, L. Sillanmäki and I. E. Holopainen, “Event-related desynchronization and synchronization during a memory task in children,” Clinical Neurophysiology, vol. 112, no. 12, pp. 2233-2240, 2001. 

[12] M. Takemi, T. Maeda, Y. Masakado, H. R. Siebner and J. Ushiba, “Muscle-selective disinhibition of corticomotor representations using a motor imagery-based brain-computer interface,” Neuroimage, vol. 183, pp. 597-605, 2018. 

[13] M. Takemi, Y. Masakado, M. Liu and J. Ushiba, “Sensorimotor event-related desynchronization represents the excitability of human spinal motoneurons,” Neuroscience, vol. 297, pp. 58-67, 2015. 

[14] D. Planelles, E. Hortal, E. Iáñez, Á. Costa and J. M. Azorín, “Processing eeg signals to detect intention of upper limb movement,” Biosystems & Biorobotics, vol. 7, pp. 655-663, 2014. 

[15] P. Francuz and E. Zabielska-Mendyk, “Does the brain differentiate between related and unrelated cuts when processing audiovisual messages? An ERP study,” Media Psychology, vol. 16, no. 4, pp. 461-475, 2013. 

[16] K. S. Heimann, S. Uithol, M. Calbi, M. A. Umiltà, M. Guerra and V. Gallese, ““Cuts in Action”: A High‐Density EEG Study Investigating the Neural Correlates of Different Editing Techniques in Film,” Cognitive Science, vol. 41, no. 6, pp. 1-34, 2016. 

[17] K. S. Heimann, M. A. Umiltà, M. Guerra and V. Gallese, “Moving mirrors: A high-density EEG study investigating the effect of camera movements on motor cortex activation during action observation,” Journal of cognitive neuroscience, vol. 26, no. 9, pp. 2087-2101, 2014. 

[18] A. Ben-Yakov and R. Henson, “The hippocampal film-editor: sensitivity and specificity to event boundaries in continuous experience,” Journal of Neuroscience, vol. 38, no. 47, pp. 10057-10068, 2018. 

[19] M. Silva, C. Baldassano and L. Fuentemilla, “Rapid memory reactivation at movie event boundaries promotes episodic encoding,” BioRxiv, vol. 39, no. 43, pp. 8538-8548, 2019. 

[20] W. Klimesch, “EEG alpha and theta oscillations reflect cognitive and memory performance: a review and analysis,” Brain research reviews, vol. 29, no. 2-3, pp. 169-195, 1999. 

[21] J. O'Keefe and M. L. Recce, “Phase relationship between hippocampal place units and the EEG theta rhythm,” Hippocampus, vol. 3, no. 3, pp. 317-330, 1999. 

[22] P. R. Shirvalkar, P. R. Rapp and M. L. Shapiro, “Bidirectional changes to hippocampal theta–gamma comodulation predict memory for recent spatial episodes,” Proceedings of the National Academy of Sciences, vol. 107, no. 15, pp. 7054-7059, 2010. 

[23] O. Jensen and C. D. Tesche, “Frontal theta activity in humans increases with memory load in a working memory task,” European journal of Neuroscience, vol. 15, no. 8, pp. 1395-1399, 2002.

---

## [Decision Letter · Decision Letter 1]

29 Sep 2021

Neural Responses to Shot Changes by Cut in Cinematographic Editing: An EEG (ERD/ERS) Study

PONE-D-21-20109R1

Dear Dr. Sanz-Aznar,

We’re pleased to inform you that your manuscript has been judged scientifically suitable for publication and will be formally accepted for publication once it meets all outstanding technical requirements.

Kind regards,

Dragan Hrncic

Academic Editor

PLOS ONE

Additional Editor Comments (optional):

Reviewers' comments:

Reviewer's Responses to Questions

**Comments to the Author**

1. If the authors have adequately addressed your comments raised in a previous round of review and you feel that this manuscript is now acceptable for publication, you may indicate that here to bypass the “Comments to the Author” section, enter your conflict of interest statement in the “Confidential to Editor” section, and submit your "Accept" recommendation.

Reviewer #1: All comments have been addressed

2. Is the manuscript technically sound, and do the data support the conclusions?

Reviewer #1: Yes

3. Has the statistical analysis been performed appropriately and rigorously? 

Reviewer #1: Yes

4. Have the authors made all data underlying the findings in their manuscript fully available?

Reviewer #1: Yes

5. Is the manuscript presented in an intelligible fashion and written in standard English?

Reviewer #1: Yes

6. Review Comments to the Author

Reviewer #1: The authors have addressed all my comments in an adequate manner, and I have only one last minor suggestion to make regarding the manuscript: in some figures the time units are denoted as 'miliseconds' or 'milisegundos', I suggest to change this for 'milliseconds' or 'ms'.

7. PLOS authors have the option to publish the peer review history of their article (what does this mean?). If published, this will include your full peer review and any attached files.

Reviewer #1: No

---

## [Editor Report · Acceptance letter]

6 Oct 2021

PONE-D-21-20109R1 

Neural Responses to Shot Changes by Cut in Cinematographic Editing: An EEG (ERD/ERS) Study 

Dear Dr. Sanz-Aznar:

I'm pleased to inform you that your manuscript has been deemed suitable for publication in PLOS ONE. Congratulations! Your manuscript is now with our production department. 

Kind regards, 

on behalf of

Professor Dragan Hrncic 

Academic Editor

PLOS ONE